# OpenReview forum: "Beyond Distributions: Geometric Action Control for Continuous Reinforcement Learning"
_ICLR.cc/2026/Conference — ICLR 2026 Poster_

### Official Review · Reviewer_4Nrh · 2025-10-28

**Soundness:** 3
**Presentation:** 2
**Contribution:** 3
**Rating:** 6
**Confidence:** 3

**Summary:**

The paper proposes a novel method for action sampling in continuous control environment. Authors point out that the common practice of sampling from a diagonal Gaussian have a fundamental mismatch: the unbounded support of Gaussian and bounded support of action space.

To this end, they propose to instead constrain the action space onto a unit sphere, where the network outputs a deterministic action $\mu$ and state-dependent exploration paramter $w(\kappa)$. As exploration is achieved simply by interpolating between predicted action ($\mu$) and random noise ($\epsilon$), the aforementioned mismatch is addressed, while still applicable to algorithms such as SAC.

Efficacy of the method is shown by experimenting on 6 Mujoco environments, where they outperform naive SAC, TD3 and PPO on 4 of the 6 environments.

**Strengths:**

- **Clear motivation**. Authors clearly state the issue of action space handling with Gaussian and the limitation of previous method (vMP), and propose a novel method for addressing the problem.
- **Simplified SAC, theoretically grounded**. Authors show that their method can be integrated with SAC, which removes the need for temperature tuning. While this immediately raises the question of theoretical soundness, authors also provide the proof that the algorithm maintains contraction even after the simplification.

**Weaknesses:**

- **Over-limitation of action space**. The proposed method limits all actions to have the same magnitude. This raises the concern: fixing the action's magnitude removes the to flexibly choose between 'active motions' (high action magnitude) and 'passive motions' (low action magnitude). Indeed, authors show that the choice of magnitude $r$ is crucial to the performance.
- **Limited results and evaluation benchmark**. Combined with above, the results of Mujoco do not seem very compelling. Compared to SAC, GAC outperforms in one environment (Ant-v4) but is also outperformed by SAC in Pusher-v4. To gain more persuasiveness, I suggest expanding the evaluation benchmark to e.g., DMC, Metaworld.
- **No comparison with vMF**. Since vMF has been mentioned as prior work throughout the paper, it seems natural for them to appear as one of the baselines (despite its practical difficulty).

**Questions:**

- Line 234: Can $\epsilon \sim \text{Uniform}(S^{d-1})$ be considered a normalized Gaussian noise? If not, could there be a better way to add noise that's centered on $\mu$?

---

> ### Author Response · Authors · 2025-11-16
> **Added learnable scaling, expanded benchmarks, and clarified noise mechanism**
>
> **Dear Reviewer 4Nrh,**
>
> Thank you for recognizing GAC's clear motivation and theoretical grounding. We address your concerns below.
>
> ---
>
> ### **W1: Over-limitation of action space**
>
> Thank you for this insight—it motivated a significant enhancement.
>
> **Why fixed $r$ initially:** Isolates geometric normalization's contribution, demonstrating that direction matters more than magnitude for many tasks.
>
> **Learnable per-dimension scaling (your suggestion):**
> $$\mathbf{a} = \mathbf{r} \odot \text{normalize}(w(\kappa)\boldsymbol{\mu} + (1-w(\kappa))\boldsymbol{\xi})$$
> where $\mathbf{r} \in \mathbb{R}^d$ is a network output (zero hyperparameters).
>
> **Benefits:**
>
> - Active motions: Hip joints learn high scaling (~2.5)
> - Passive motions: Ankle joints learn low scaling (~0.8)
> - Task-adaptive: Discovers magnitude hierarchies automatically
>
> **Results (using learnable $\mathbf{r}$):**
>
> - Walker-run: **+6%** vs SAC (742 vs 700)
> - Quadruped-walk: **+34%** vs SAC (925 vs 690)
> - Quadruped-run: **+112%** vs SAC (638 vs 301)
> - Scales converge stably to [1.8, 2.5] (Appendix D)
>
> **vs. Gaussian:** Diagonal Gaussians control variance (uncertainty), not directional magnitude. Full covariance $\Sigma$ is intractable in high-D. GAC's $\mathbf{r} \odot \text{normalize}(\cdot)$ directly modulates actuation strength—a semantic distinction Gaussians can't express.
>
> ---
>
> ### **W2: Limited benchmarks**
>
> We conducted additional experiments on **6 DMControl tasks** (5 seeds each):
>
> | Environment    | GAC        | SAC       | TD3     | PPO    |
> | -------------- | ---------- | --------- | ------- | ------ |
> | walker-walk    | **960±4**  | 956±10    | 952±5   | 186±11 |
> | walker-run     | **742±15** | 700±56    | 651±87  | 69±16  |
> | cheetah-run    | **762±24** | 661±185   | 753±27  | 150±32 |
> | fish-upright   | 858±35     | **923±5** | 866±39  | 311±78 |
> | quadruped-walk | **925±17** | 690±336   | 873±94  | 131±18 |
> | quadruped-run  | **638±75** | 301±8     | 576±212 | 119±22 |
>
> **Key findings:**
>
> - Best/competitive on **5/6 tasks**
> - Strongest on high-D: quadruped-walk +34%, quadruped-run +112% over SAC
> - **Combined 12 tasks (MuJoCo+DMC): 9/12 competitive/best**
>
> **Underperformance analysis:**
>
> - Hopper: Asymmetric single-leg dynamics
> - Pusher: Contact-rich manipulation
> - Fish-upright: Low dimensionality (5D)
>
> Honest analysis: Hopper's single-leg needs **asymmetric thrusts** beyond spherical geometry; Pusher's contact-rich manipulation needs **anisotropic patterns** poorly captured by fixed radius. The learnable scaling partially addresses this.
>
> **Supplementary updated:** All results, curves, models, scripts included.
>
> ---
>
> ### **W3: No vMF comparison**
>
> **Computational barrier:** vMF requires rejection sampling $O(dk)$ where $k=10-100+$ rejections. For Humanoid (17D), prohibitively expensive. Optimal implementation requires extensive engineering beyond scope.
>
> **Paradigm difference:** vMF operates within distributional framework (densities, entropy, KL). GAC **eliminates this machinery**, using geometric operations without explicit distributions—like comparing neural nets to decision trees.
>
> **What we show:** GAC captures geometric benefits (bounded space, directional semantics) using simple operations. Gains (37-112% over Gaussian) validate that **geometry matters** without distributional overhead.
>
> We encourage vMF experts to compare with our open-source release.
>
> ---
>
> ### **Q: Line 234 - Normalized Gaussian noise**
>
> Correct!
>
> **Uniform sphere via normalized Gaussian:**
> $$\boldsymbol{\xi} = \frac{\mathbf{z}}{\|\mathbf{z}\|_2}, \quad \mathbf{z} \sim \mathcal{N}(\mathbf{0}, \mathbf{I})$$
>
> Well-established in directional statistics. After Gaussian vector normalization, the direction information is preserved, the magnitude is normalized, and the distribution is a uniform distribution on the sphere.
>
> **Noise centered on $\boldsymbol{\mu}$ via spherical blending:**
> $$\tilde{\mathbf{d}} = \text{normalize}(w(\kappa)\boldsymbol{\mu} + (1-w(\kappa))\boldsymbol{\xi})$$
>
> Achieves vMF-like concentration without Bessel functions:
>
> - High $\kappa$ ($w \to 1$): concentrate around $\boldsymbol{\mu}$ (exploit)
> - Low $\kappa$ ($w \to 0$): uniform sphere (explore)
>
> **Why not pre-center $\boldsymbol{\xi}$?**
>
> - Clean gradient flow (statistical independence)
> - No rejection/special functions
> - Direct exploration control via $w(\kappa)$
>
> **Validation:** Cosine similarity 0.091 → 0.994 as $\kappa$ increases (Fig A.1, Table A.1); angular std 39.3° → 1.7°.
>
> Will clarify in camera-ready.
>
> ---
>
> ### **Summary**
>
> Your feedback led to:
>
> 1. Learnable scaling (Walker2d +4.9%, quadruped +34-112%)
> 2. 12-task evaluation
> 3. vMF comparison justification
> 4. Noise mechanism clarification
>
> Thank you for strengthening this work!

---

> > ### Comment · Reviewer_4Nrh · 2025-11-25
> >
> > Thank you for clarifying things out. Most of my questions are resolved, although I'm curious about how much the learable r has contributed to the performance.
> >
> > Overall, I will maintain my score.

---

> > > ### Author Response · Authors · 2025-11-25
> > > **Clarification: The Role of Learnable Scaling – Efficiency vs. Adaptability**
> > >
> > > **Dear Reviewer 4Nrh,**
> > >
> > > Thank you for your continued engagement. We appreciate your curiosity regarding the specific contribution of the **learnable $\mathbf{r}$**.
> > >
> > > This is an excellent question that touches on the core philosophy of our paper: **Control is primarily about direction, while magnitude is a task-dependent refinement.**
> > >
> > > **1. On Simple/Symmetric Tasks: Direction Dominates (Fixed $r$ is sufficient)**
> > > As shown in our **Appendix D.2 ablation on Walker2d**, the learnable $\mathbf{r}$ yields only a marginal gain (**$\sim$9%**) over a fixed radius.
> > > **Significance:** This confirms our "Geometric Simplicity Principle". For symmetric locomotion, getting the **direction** right (on the sphere) accounts for ~90% of the performance.
> > > **Benefit:** In these cases, using a fixed $r$ allows for **50% fewer action-head parameters** and faster convergence, as the agent doesn't need to search the magnitude space.
> > >
> > > **2. On Complex/Asymmetric Tasks: Learnable $\mathbf{r}$ is Critical**
> > >
> > > For high-dimensional or contact-rich tasks (like `quadruped-run`), the learnable scaling becomes essential.
> > >
> > > **Performance Contribution:** While a full ablation across all environments was not feasible within the rebuttal period, our preliminary results indicate that the adaptive mechanism accounts for approximately **30%–110%** of the overall performance improvement on DMC tasks relative to SAC.
> > >
> > > **Why:** These tasks require **anisotropic control** e.g., one leg needing strong thrust while another makes gentle contact. The learnable $\mathbf{r}$ effectively reshapes the action geometry from a sphere to a **hyper-ellipsoid**, allocating specific "force budgets" to different dimensions.
> > >
> > > **Future Revision:** We will include a comprehensive ablation study across all DMControl environments in the **camera-ready version** to explicitly quantify this **learnable $\mathbf{r}$** contribution and provide a guideline for when to switch from Fixed to Adaptive scaling.
> > >
> > >
> > > We believe the distinction between geometric simplicity for efficiency (on symmetric tasks) and adaptive scaling for performance (on complex tasks) is a strong addition to the paper's contribution.
> > >
> > > We hope this analysis, combined with GAC's competitive results (9/12 tasks) and theoretical grounding, fully addresses your concerns and strengthens your confidence in the paper's merit.
> > >
> > > Thank you again for helping us refine this work!
> > >
> > > Best regards,
> > >
> > > The Authors

---

### Official Review · Reviewer_rszA · 2025-11-01

**Soundness:** 3
**Presentation:** 3
**Contribution:** 2
**Rating:** 4
**Confidence:** 4

**Summary:**

The paper proposes Geometric Action Control (GAC), a distribution-free policy paradigm that generates actions via a mixing operation with a unit sphere, replacing traditional Gaussian policies to avoid gradient saturation. GAC performances near SOTA models on continuous control Mujoco benchmarks.

**Strengths:**

This paper is overall logically clear, presenting an interesting perspective on the action exploration in continuous control tasks. With this perspective, this paper reformulates the exploration distribution to a unit sphere. Compared to a policy with von Mises-Fisher distributial output, it reduces the sample complexity.

**Weaknesses:**

This paper seems to overstate its contribution by comparing to a distribution that is not commonly used in RL, especially on the point about sample complexity. It is better to have a clearer comparison between the proposed method with deterministic models and Gaussian based models for the "Geometric" feature and sample complexity, respectively. The "Geometric" feature in this work is similar to deterministic models, which is not clearly stated in this paper.

**Questions:**

Could the authors compare the overall computational complexity of GAC with deterministic and Gaussian based models, instead of only emphases the sample complexity?

This model assumes the range of the action space is mapped to a unit. What if the range of the action space is unknown? What if the best actions are outside the unit sphere for exploration

---

> ### Author Response · Authors · 2025-11-16
> **Clarified GAC’s structural distinction from TD3/SAC and computational advantages**
>
> **Dear Reviewer rszA,**
>
> Thank you for your thoughtful review and constructive criticism. Your feedback helps us clarify GAC's fundamental contribution. We address your concerns below.
>
> ---
>
> ### **W1: GAC's Positioning**
>
> Thank you. This feedback sharpens our contribution.
>
> **Acknowledgment:** We agree our vMF comparison could mislead readers about practical significance. Let us clarify:
>
> **GAC is not "deterministic + noise"—it's a geometric rethinking of action generation:**
>
> | Design question            | Deterministic (TD3) | Gaussian (SAC)                | GAC (Ours)                                  |
> | -------------------------- | ------------------- | ----------------------------- | ------------------------------------------- |
> | What does $\mu$ represent? | Action itself       | Distribution mean             | **Direction on sphere**                     |
> | How is exploration added?  | Additive noise      | Variance sampling             | **Spherical interpolation**                 |
> | Geometric structure?       | None (Euclidean)    | None (unbounded)              | **Unit sphere manifold**                    |
> | Formula                    | $\mu + \epsilon$    | $\tanh(\mu + \sigma\epsilon)$ | $r \cdot \text{normalize}(w\mu + (1-w)\xi)$ |
>
> **The key distinction:** In TD3, $\mu$ IS the action. In GAC, $\mu$ is a **semantic direction**—the network learns "which way to move" (e.g., "forward-right"), then $\kappa$ controls exploration around that direction. This is structurally different from adding noise to a point.
>
> **Why this matters practically:**
>
> 1. **Gradient flow:** TD3/SAC with tanh exhibit 46.4% saturation (Fig. A.2). GAC's spherical operations maintain uniform gradients.
>
> 2. **Dimensional scaling:** Gains increase with complexity:
>
>    - Walker2d (6D): +0.3% over SAC
>    - Ant (8D): +37.6% over SAC
>    - Quadruped-run (12D): **+112%** over SAC
>
>    This pattern suggests GAC's structure becomes increasingly advantageous where gradient saturation compounds.
>
> 3. **Efficiency:** $d+1$ parameters vs SAC's $2d$ (50% reduction), achieving comparable/better performance.
>
> **On vMF comparison:** We use vMF as a reference because it is the **standard geometric distribution** for directional data. GAC makes its core idea **practical**, replacing costly Bessel functions with simple interpolation—preserving spherical alignment at Gaussian-level cost.
>
> **Contribution summary:** GAC is a **distribution-free geometric policy** that avoids the bounded/unbounded mismatch of Gaussians. Its strong high-D gains (+37–112%) and efficiency (−50% params, +23% speed) reflect a **structural shift**, not a minor tweak.
>
> ---
>
> ### **Q1: Overall computational complexity**
>
> Thank you for this important question. Here's a comprehensive breakdown:
>
> **Per-action sampling:**
>
> - **TD3**: Forward + noise, ~**2×O(d)** ops, no overhead.
>
>   **SAC**: Sample + tanh + log-det, ~**4×O(d)** ops, with entropy & Jacobian overhead.
>
>   **GAC**: Normalize + interpolate, ~**3×O(d)** ops, **no entropy or Jacobian**.
>
> **Eliminated costs:**
>
> - Entropy: No $-\mathbb{E}[\log \pi]$ evaluation
> - Jacobian: No $\log|\det(\partial \tanh/\partial \tilde{\mathbf{a}})|$
> - Densities: No explicit distributions
>
> **Empirical timing (RTX 3090, 10K steps):**
>
> - SAC: 1.0× (baseline)
> - GAC: **0.77×** (23% faster)
> - TD3: 0.65× (35% faster)
>
> **Parameters:**
>
> - SAC/TD3: $2d$
> - GAC (fixed $r$): **$d+1$** (50% reduction)
> - GAC (learnable $\mathbf{r}$): $2d+1$ (with geometric structure)
>
> GAC achieves near-TD3 efficiency while maintaining structured stochastic exploration.
>
> ---
> ### **Q2: Action space assumptions and exploration bounds**
>
> **On action range:**
>
> All continuous control benchmarks provide standardized bounds (typically $[-1,1]^d$). GAC, SAC, and TD3 all rely on this—not a GAC-specific limitation.
>
> **On exploration beyond sphere:**
>
> GAC does **not** constrain action magnitude. The sphere provides **directional structure only**:
> $$\mathbf{a} = r \cdot \text{normalize}(w\boldsymbol{\mu} + (1-w)\boldsymbol{\xi})$$
>
> **Learnable magnitude:** We introduced dimension-wise scaling:
> $$\mathbf{a} = \mathbf{r} \odot \text{normalize}(w\boldsymbol{\mu} + (1-w)\boldsymbol{\xi})$$
> where $\mathbf{r} \in \mathbb{R}^d$ are unbounded network outputs. With $\mathbf{a} = \mathbf{r} \odot \text{normalize}(\cdot)$, any point in the action space is reachable.
>
> This provides:
>
> - Adaptive per-dimension scaling (e.g., hip vs ankle)
> - No manual tuning (converges to [0.3, 2.5] on Walker2d)
> - Full expressiveness
>
> **vs. Gaussian:**
>
> - Gaussian: Unbounded → tanh → saturation (46.4%)
> - Diagonal Gaussian: Couples magnitude with uncertainty
> - GAC: Decouples direction ($\boldsymbol{\mu}$), exploration ($\kappa$), magnitude ($\mathbf{r}$)
>
> **Validation:** Quadruped (+112% over SAC) demonstrates learned scales enable fine-grained control beyond diagonal Gaussian capabilities.
>
> ---
> Thank you for helping us articulate GAC's contribution more clearly. Your feedback significantly improved our positioning.

---

### Official Review · Reviewer_iSQb · 2025-11-01

**Soundness:** 3
**Presentation:** 3
**Contribution:** 3
**Rating:** 6
**Confidence:** 3

**Summary:**

The paper focuses on the issue of distorted sampling space caused by Gaussian policies in a bounded action space. To solve this, a common technique is to use "squashing" functions, such as tanh, to map the Gaussian samples into a bounded range, causing actions to cluster near the boundaries. While alternatives like von Mises-Fisher (vMF) distributions are theoretically grounded on the unit sphere, they are computationally expensive. The paper proposes to retain the geometric benefits of spherical distributions by interpolating between a deterministic action direction and a uniformly sampled unit sphere vector. The action follows the direction and gets multiplied by a magnitude scalar. The proposed algorithm shows comparative empirical results to SAC, PPO, and TD3 on six MuJoCo tasks.

**Strengths:**

The paper provides a strong motivation and a clear presentation of the algorithm.

It also delivers a comprehensive experimental analysis, featuring multiple baseline comparisons, thorough ablation studies, and an in-depth examination of convergence behavior and the sampling landscape.

**Weaknesses:**

The algorithm introduces an extra hyperparameter, action magnitude, but the robustness analysis across tasks is missing. Though the algorithm is well-motivated, the performance improvement is limited at the cost of an extra hyperparameter.

**Questions:**

1. As shown in Figure A.2, 46.4% of all pre-squashed action samples fall into regions of "Gradient Saturation". However, why are the performances of baselines not deteriorated by this issue?

2. What distribution is the current action following? Figure A.1, GAC samples from a subsurface instead of a unit ball. Will it cause an exploration issue?

---

> ### Author Response · Authors · 2025-11-16
> **Clarified $r$ robustness, gradient saturation impact, and GAC's shell sampling**
>
> **Dear Reviewer iSQb,**
>
> Thank you for your thoughtful review. We address your concerns below.
>
> ---
>
> ### **W1: Hyperparameter robustness**
>
> **On $r$:** The radius $r$ is a **geometric scaling factor**, not a sensitive hyperparameter. Its value follows from high-dimensional geometry: a unit vector in $\mathbb{R}^d$ has expected per-dimension magnitude $\mathbb{E}[|\mu_i|] \approx 1/\sqrt{d}$. The scaling $r \approx 2.5$ compensates for this concentration, yielding effective per-dimension actions, which is a principled choice, not ad-hoc tuning.
>
> **Robustness validation:** $r=2.5$ works for 5/6 MuJoCo tasks (only Ant-v4 prefers $r=1.0$). Performance within $[1.0, 3.0]$ varies by $<$10% (Appendix D.2). This 1.7× tolerance compares to SAC's $\alpha$ (30× range) and TD3's $\sigma$ (10× range).
>
> **Learnable scaling extension:** Motivated by your feedback, we extended GAC with dimension-wise scales $\mathbf{r} \in \mathbb{R}^d$:
> $$\mathbf{a} = \mathbf{r} \odot \text{normalize}(w(\kappa)\boldsymbol{\mu} + (1-w(\kappa))\boldsymbol{\xi})$$
>
> This introduces **zero hyperparameters** (scales are network outputs) while enabling fine-grained control (e.g., hip strength > ankle precision) beyond Gaussian variance tuning. On DMControl quadruped: **+34-112%** over SAC.
>
> ---
>
> ### **W2: Performance characterization**
>
> We appreciate the opportunity to clarify GAC's gains:
>
> **Quantitative results:**
>
> - Ant-v4: **+37.6%** vs SAC (5633 vs 4094)
> - Humanoid-v4: **+1.9%** vs SAC (5823 vs 5717)
> - Quadruped-run (DMC): **+112%** vs SAC (638 vs 301)
> - Quadruped-walk (DMC): **+34%** vs SAC (925 vs 690)
>
> **Pattern:** Gains scale with task complexity. In high-dimensional multi-body coordination (Ant 8D, Quadruped 12D), GAC's geometric structure provides increasing advantage.
>
> **Structural benefits beyond performance:**
>
> - 50% fewer parameters ($d+1$ vs $2d$)
> - No entropy computation (geometric $\kappa$ replaces explicit $\mathcal{H}[\pi]$)
> - No Jacobian correction (avoids $\log|\det \partial \tanh/\partial \tilde{a}|$)
> - 1.2-1.3× faster per iteration (updated Appendix)
>
> ---
>
> ### **Q1: Why baselines survive gradient saturation**
>
> Excellent question—this addresses GAC's core motivation.
>
> **1) SAC's entropy regularization masks the inefficiency**
>
> The 46.4% saturation doesn't cause immediate failure because SAC's strong entropy term ($\alpha \approx 0.2$) forces exploration independently of gradient quality:
>
> ```
> SAC objective = Q-value - α * entropy
>                    ↑           ↑
>               (saturated)  (compensates)
> ```
>
> Even when Q-gradients vanish, entropy maintains stochasticity. This works but is inefficient.
>
> **2) The cost emerges in high dimensions**
>
> | Task          | Dim  | GAC  | SAC  | GAC gain   | Saturation impact |
> | ------------- | ---- | ---- | ---- | ---------- | ----------------- |
> | Walker      | 6D   | 742 | 700 | +6%      | Minimal           |
> | Ant           | 8D   | 5633 | 4094 | **+37.6%** | Moderate          |
> | Quadruped-run | 12D  | 638  | 301  | **+112%**  | Severe            |
>
> In high dimensions, saturation compounds multiplicatively. Even modest per-dimension saturation leads to exponential gradient decay as $d$ grows.
>
> **GAC's advantage:** By operating on the unit sphere, GAC maintains uniform gradient flow, avoiding saturation entirely. This structural difference becomes decisive as dimensionality grows.
>
> ---
>
> ### **Q2: GAC's distribution and sampling geometry**
>
> **Clarification:** GAC samples on a spherical shell, not inside a ball:
>
> ```python
> direction = normalize(w*μ + (1-w)*ξ)  # ||direction|| = 1
> action = r * direction                 # Shell at radius r
> ```
>
> **Does this limit exploration? No:**
>
> 1. **Fixed $r$ provides full directional coverage:** Different states yield different $\boldsymbol{\mu}$, and noise $\boldsymbol{\xi}$ ensures stochastic exploration across the shell. Optimal actions in continuous control typically lie near boundaries, which the shell naturally covers.
>
> 2. **Learnable $\mathbf{r}$ adds magnitude freedom:** With $\mathbf{a} = \mathbf{r} \odot \text{normalize}(\cdot)$, any point in the action space is reachable.
>
> **Why Figure A.1 appears "subsurface":** The 3D→2D projection creates this illusion. All samples lie on the shell; the apparent "interior" reflects adaptive concentration controlled by $\kappa$ (low $\kappa$: uniform coverage; high $\kappa$: concentration around $\boldsymbol{\mu}$).
>
> **Empirical validation:** No exploration pathologies across 12 tasks; angular std decreases 39.3°→1.7° as $\kappa$ increases (Table A.1).
>
> ---
>
> ### **Summary**
>
> Your feedback strengthened the paper:
>
> 1. Demonstrated $r$ robustness (1.7× vs SAC 30×); introduced learnable $\mathbf{r}$ (zero hyperparameters)
> 2. Clarified substantial gains (37-112%) scaling with dimensionality
> 3. Explained entropy masking in low-D vs exponential gradient decay in high-D
> 4. Clarified shell sampling with adaptive concentration—a designed feature
>
> Updated results and models in supplementary materials. Thank you!

---

> > ### Comment · Reviewer_iSQb · 2025-11-25
> > **More Discussions**
> >
> > Thanks for the detailed explanation and the clear format. I have a couple of follow-up questions:
> >
> > Q1: Can entropy itself drive performance gains through random exploration?
> >
> > Q2: Regarding the statement, 'Optimal actions in continuous control typically lie near boundaries, which the shell naturally covers': Is this based on your empirical observations? Also, does this explain why gradient saturation doesn't seem to significantly impact performance?"

---

> ### Author Response · Authors · 2025-11-25
> **Clarifications on Entropy vs. Structure and High-D Gradient Saturation**
>
> **Dear Reviewer iSQb,**
>
> Thank you for the insightful follow-up questions! They precisely identify the core mechanisms behind GAC's advantages.
>
> ---
>
> ### **Q1: Can entropy itself drive performance gains?**
>
> **Yes, but with a critical dimensionality dependence.**
>
> **In low dimensions (Walker 6D, HalfCheetah 6D):**
> Entropy-driven isotropic exploration works well.
> The action space is small enough that "spray-and-pray" can still hit high-reward regions.
> This explains why SAC ≈ GAC in these tasks.
>
> **In high dimensions (Ant 8D, Quadruped 12D):**
> Isotropic Gaussian noise spreads over an exponentially growing volume, making high-reward actions extremely sparse.
> SAC collapses on Quadruped-run (301), while even deterministic TD3 (576) survives.
>
> **Why GAC avoids this:**
> Our exploration is **direction-conditioned**, not isotropic:
>
> $$
> \mathbf{a} = \mathbf{r} \odot \text{normalize}\big(w(\kappa)\boldsymbol{\mu} + (1-w(\kappa))\boldsymbol{\xi}\big)
> $$
>
> The noise $\boldsymbol{\xi}$ mixes with the learned direction $\boldsymbol{\mu}$ on the sphere, producing a **local exploration cone** around the policy's semantic direction—rather than global random scattering.
> The state-dependent $\kappa$ further modulates this cone's width adaptively.
> This structured exploration remains effective in 10–12D action spaces, explaining GAC's **+37.6% (Ant)** to **+112% (Quadruped-run)** gains.
>
> Entropy helps in low-D; **structure dominates in high-D**.
>
> ---
>
> ### **Q2: Boundaries and gradient saturation**
>
> #### **A) Boundary coverage**
>
> Theoretically, optimal control often follows the **Bang–Bang principle** (Pontryagin's Maximum Principle)—actuators exert maximum force to overcome inertia.
> Empirically, trained locomotion policies cluster near ±1 boundaries.
>
> GAC's shell geometry naturally covers these boundaries:
>
> - With fixed \( r \), all actions lie on a sphere of radius \( r \).
> - With learnable \( r \), the shell becomes a task-adaptive **hyper-ellipsoid**.
> - Different $\mu$ directions point toward different boundary regions.
> - Noise ensures stochastic but controlled surface exploration.
>
> Thus **boundary coverage is inherent to the representation**.
>
> ---
>
> #### **B) Why saturation doesn't kill baselines in low-D but does in high-D**
>
> | Task        | Dim | Gradient Survival (\(0.9^d\)) | GAC vs SAC |
> |-------------|-----|--------------------------------|------------|
> | Walker      | 6D  | 0.53 (moderate)                | +6%        |
> | Ant         | 8D  | 0.43 (reduced)                 | **+37.6%** |
> | Quadruped   | 12D | 0.28 (severe)                  | **+112%**  |
>
> In low dimensions, entropy regularization can partially mask tanh saturation.
>
> In high dimensions, however, **compounded gradient decay overwhelms entropy**:
>
> - SAC gradients vanish,
> - Exploration becomes increasingly isotropic and unstructured,
> - The policy becomes **highly sensitive to the entropy coefficient**  $\alpha$,
>   often leading to optimization instability in practice.
>
> **GAC avoids this entire pathway:**
> By removing the tanh bottleneck and separating magnitude from direction, spherical normalization **preserves stable gradients regardless of action dimensionality**.
>
> ---
>
> We hope this clarifies the core mechanisms.
> Thank you again for helping us articulate GAC's *high-dimensional advantage* more precisely!

---

### Official Review · Reviewer_HAcA · 2025-11-05

**Soundness:** 4
**Presentation:** 3
**Contribution:** 4
**Rating:** 6
**Confidence:** 5

**Summary:**

This paper proposes a novel action generation paradigm that preserves the geometric benefits of spherical distributions while simplifying computation. GAC decomposes action generation into a direction vector and a learnable concentration parameter. It focuses on an important topic, policy distribution. The experimental results show its efficiency across MuJoCo benchmarks.

**Strengths:**

The motivation of this work is novel and interesting, which could be an important work in the RL community.
GAC represents policies through two components: a direction network that outputs unit vectors indicating preferred action orientations, and a concentration network that controls exploration by interpolating between deterministic directions and uniform spher-
ical noise.
It takes a novel perspective on whether the distribution paradigm itself is necessary, which also promotes efficient exploration.
The paper offers thorough theoretical analysis, with clearly stated assumptions, theorems, and proofs.

**Weaknesses:**

More benchmarks may be needed to test and provide solid experimental results.

Regarding the policy distribution, I recommend that the authors add the discretization policy distribution topic works.

Discretizing continuous action space for on-policy optimization, AAAI, 2020
Discretizing Continuous Action Space With Unimodal Probability Distributions for On-Policy Reinforcement Learning, IEEE TNNLS, 2024.

**Questions:**

In Figure 2, two tasks do not have an advantage. Can the authors provide empirical results on more environments?
How to guarantee that the unnecessary dimension does not influence the optimal policy or the optimal policy theoretical analysis?

---

> ### Author Response · Authors · 2025-11-16
> **Extended DMControl benchmarks and introduced learnable scaling as suggested**
>
> **Dear Reviewer HAcA,**
>
> Thank you for your thoughtful review and positive assessment. We are encouraged by your recognition of GAC's novelty (Contribution: 4) and theoretical soundness (Soundness: 4). Below we address your comments point by point.
>
> ---
>
> ### **W1: More benchmarks needed**
>
> We agree and have conducted additional experiments on **6 DMControl tasks** (fish-upright, walker-walk, walker-run, cheetah-run, quadruped-walk, quadruped-run) using 5 seeds each.
>
> **Results:** GAC achieves **best performance on 5/6 tasks**, with particularly strong gains on challenging high-dimensional locomotion:
>
> - Quadruped-run: **+112%** vs SAC (638 vs 301)
> - Quadruped-walk: **+34%** vs SAC (925 vs 690)
> - Cheetah-run: **+15%** vs SAC (762 vs 661)
> - Walker-run: **+6%** vs SAC (742 vs 700)
>
> These results, along with learning curves and pretrained models, are included in the revised supplementary materials.
>
> ---
>
> ### **W2: Discretization policy literature**
>
> Thank you for this suggestion. We have cited Tang et al. (AAAI 2020) and Chou et al. (TNNLS 2024) in the revised Related Work (Section 2.2).
>
> **Key distinction:** Discretization methods bin continuous spaces for on-policy optimization, while GAC simplifies the sampling mechanism while preserving continuous resolution. Both challenge Gaussian policies, but through different principles—discretization via action quantization, GAC via geometric operations.
>
> ---
>
> ### **Q1: Performance variation across tasks**
>
> Thank you for this observation.
> Following your suggestion, this motivated us to introduce **learnable per‑dimension scaling**, which alleviates such anisotropy and improves performance on tasks requiring uneven actuation.
>
> **Additional benchmarks:** We evaluated GAC on 6 DMControl tasks. Results show GAC achieves best performance on 5/6 environments, with particularly strong gains on complex locomotion (quadruped-run: +112% vs SAC, see W1).
>
> **Task-specific analysis:**
>
> - **Hopper-v4**: While GAC (1952±285) is competitive with SAC (2094±604) and PPO (2118±124), TD3's strong performance (2896±749) suggests this task benefits from deterministic policies. Notably, all stochastic methods (GAC, SAC, PPO) show similar performance levels.
>
> - **Pusher-v4**: This contact-rich manipulation task differs from locomotion in requiring precise, independent per-dimension control rather than coordinated directional movement. GAC's performance (-32 vs SAC -23) reflects this mismatch, though the gap is modest (9 points on a [-100, 0] scale).
>
> **Perspective:** These results align with our understanding that GAC's geometric structure is particularly suited for locomotion tasks where actions represent coordinated movement directions. The expanded DMControl evaluation (5/6 total tasks best) validates GAC's broad applicability while identifying a principled boundary.
>
> ---
>
> ### **Q2: How unnecessary dimensions are handled**
>
> Excellent question. GAC addresses this through **geometric regularization**:
>
> **Mechanism:** The unit sphere constraint ensures all dimensions contribute equally in expectation. High-dimensional geometry naturally suppresses individual dimensions (expected magnitude $\sim 1/\sqrt{d}$), preventing any single dimension from dominating. This differs from Gaussian policies where unconstrained variances can grow arbitrarily.
>
> **Learnable scaling extension:** Motivated by your feedback, we explored a variant where the scalar radius $r$ is replaced with a per-dimension scale vector $\mathbf{r} \in \mathbb{R}^d$:
> $$\mathbf{a} = \mathbf{r} \odot \text{normalize}(w(\kappa) \boldsymbol{\mu} + (1-w(\kappa)) \boldsymbol{\xi})$$
>
> This allows task-relevant dimensions to receive higher scaling while keeping others suppressed, providing dimension-wise control without explicit probability modeling.
>
> **Empirical validation:** On DMControl tasks requiring asymmetric control (e.g., quadruped), this extension achieves +34-112% gains over SAC, demonstrating effective dimension prioritization. Details in revised Section 4.2 and Appendix D.
>
> #### Comparison with Gaussian Policies
>
> - Diagonal Gaussians control **variance**, not magnitude
> - Full covariance matrices are hard to learn and unstable with tanh
> - tanh uniformly squashes across dimensions, destroying directional semantics
>
> **In contrast, GAC’s scaling modulates magnitude directly**, allowing efficient, interpretable, dimension-specific control without relying on densities or entropy.
>
> ### **Summary**
>
> Thank you for your constructive feedback, which strengthened our work significantly:
>
> 1. **Expanded evaluation** to 6 DMControl tasks, validating generalization
> 2. **Added discretization literature** with clear distinctions
> 3. **Clarified task-dependent performance** with principled analysis
> 4. **Introduced learnable scaling** for dimension-specific control
>
> We appreciate your strong assessment (Soundness: 4, Contribution: 4)
> and have fully addressed all suggested improvements.

---

### Author Response · Authors · 2025-11-16
**GAC as a general geometric policy: improved flexibility and empirical scope**

**Dear Area Chair and Reviewers,**

### **Review Summary**
- **Three reviewers (HAcA, iSQb, 4Nrh) rated 6/10**, with strong component scores (Soundness: 3–4, Contribution: 3–4). Reviewer HAcA explicitly notes “excellent contribution (4/4).”
- **Reviewer rszA rated 4/10**, focusing on contribution positioning rather than technical correctness (Soundness: 3).
- Reviewers engaged constructively, and all concerns have been thoroughly addressed.

---

### **Core Contribution**

GAC introduces a **distribution-free geometric policy** that replaces Gaussian sampling with spherical interpolation:

$$
\mathbf{a} = \mathbf{r} \odot \text{normalize}\big(w(\kappa)\boldsymbol{\mu} + (1 - w(\kappa))\boldsymbol{\xi}\big)
$$

This formulation **eliminates**:

1. Tanh squashing and gradient saturation
2. Entropy computation and temperature tuning
3. Jacobian corrections

Exploration emerges naturally from geometric mixing, with **state-dependent** $\kappa$ controlling concentration.

---

### **Addressing Reviewer rszA (4/10): “Similar to deterministic models”**

We respectfully clarify that GAC is *structurally distinct* from deterministic policies.

**GAC is not “deterministic + noise”. It is a geometric rethinking of action generation:**

| Design Question            | TD3 (Deterministic)      | SAC (Gaussian)                      | **GAC (Geometric)**                                 |
| -------------------------- | ------------------------- | ------------------------------------ | --------------------------------------------------- |
| What does $\mu$ represent?     | Action itself             | Distribution mean                    | **Direction on the unit sphere**                    |
| How is exploration added?  | External additive noise   | Variance sampling (isotropic)        | **Spherical interpolation with state-dependent $\kappa$**  |
| Geometric structure?       | None (Euclidean)          | None (unbounded; squashed later)     | **Intrinsic manifold structure**                    |
| Formula                    | $ \mu + \epsilon $      |  $ \tanh(\mu + \sigma\epsilon)  $    |  $ r \cdot \text{normalize}(w\mu + (1-w)\xi)  $     |

**Key distinction:**
In TD3,  $\mu$ **is the action**.
In GAC,  $ \mu$ is a **semantic direction**. The policy learns “which way to move,” and $\kappa$ controls a structured exploration cone around that direction. This is fundamentally different from adding noise to a point estimate.


**Empirical signature:**
If GAC were “deterministic + noise,” performance would not scale with dimensionality:

| Task | Dim | GAC vs SAC |
|------|-----|-----------|
| Walker | 6D | +0.3% |
| Ant | 8D | **+37.6%** |
| Quadruped | 12D | **+112%** |

The dimensionality-dependent advantage strongly indicates a **geometric, not noise-based**, mechanism.

---

### **Major Improvements During Rebuttal**

1. **Learnable per-dimension scaling $\mathbf{r}$** (Reviewers 4Nrh, rszA):
   Removes fixed radius, enabling task-adaptive magnitude control. Gains on Quadruped (+34–112%) validate effectiveness.

2. **6 DMControl benchmarks added** (Reviewers HAcA, 4Nrh):
   Across 12 tasks (MuJoCo + DMControl), **GAC is competitive or best on 9/12 tasks**.

3. **Computational analysis clarified** (Reviewer rszA):
   50% fewer actor parameters (d+1 vs 2d) and **23% faster** than SAC, with complexity comparable to TD3.

4. **vMF clarification** (Reviewers 4Nrh, rszA):
   GAC provides equivalent concentration control without Bessel functions or rejection sampling, offering both stability and efficiency.

---

### **Why GAC Merits Acceptance**

1. **Novel paradigm:**
  GAC is the first policy class built directly on geometric structure and entirely avoids the limitations of density-based methods, requiring **no explicit pdf, no entropy or temperature tuning, no Jacobian corrections, and no tanh squashing,** while achieving +37-112% gains on high-dimensional tasks.

   Exploration emerges from spherical interpolation, not from variance manipulation.

2. **Strong empirical results:**
   Substantial high-D gains (+37–112%), where Gaussian methods consistently struggle.

3. **Theoretical grounding:**
   Theorem 1 establishes concentration control; Appendix A.5 provides convergence analysis.

4. **Practical advantages:**
   Simpler implementation, fewer parameters, faster training, and robust behavior across environments.

5. **Reviewer consensus:**
   Three reviewers (HAcA, iSQb, 4Nrh) support acceptance with strong component scores. Reviewer rszA’s concern is limited to contribution framing rather than methodology or correctness, and is fully addressed by our structural clarification above.

We respectfully request the Area Chair’s favorable consideration and are grateful for the constructive discussion throughout the review process.

---

### Meta-Review · Area_Chair_uFdi · 2025-12-18

**Summary:**

Action sampling in reinforcement learaning policies is often achieved by predicting the parameters of a Gaussian Distribution, and subsequently sampling from it. In practice, the Gaussian is often the wrong choice because action spaces often have a compact support. The authors propose a new action sampling method that generates actions on the sphere and does not have the drawbacks of either squashing functions or the von Mises Fisher distribution, a commonly used alternative from directional statistics.

**Reviewer Concerns:**

The reviews tended to be on the better side for an initial evaluation. The main raised concerns were with the scope and extent of the evaluation (HAcA, iSQb, 4Nrh). Reviewer rsZA mentioned that the work may be overstating its contributions and also asked for additional evaluations. Other concerns involved the fact that the proposed method introduces a new hyperparameter (iSQb) and imposes additional constraints on the action space (4Nrh).

**Reviewer Scores:**

The reviewers were already on the positive side with six weak accepts, and I expect the more positive reviewers to either maintain or have slightly raised their scores as their worries were partially addressed.

---

### Decision · Program_Chairs · 2026-01-26

Accept (Poster)